# Single-Electron Transport and Detection of Graphene Quantum Dots

**DOI:** 10.3390/nano13050889

**Published:** 2023-02-27

**Authors:** Xinxing Li, Jinggao Sui, Jingyue Fang

**Affiliations:** 1School of Physics and Electronics, Central South University, Changsha 410083, China; 2Key Laboratory of Nanodevices, Suzhou Institute of Nano-Tech and Nano-Bionics, CAS, Suzhou 215213, China; 3National Innovation Institute of Defense Technology, Academy of Military Sciences PLA China, Beijing 100071, China

**Keywords:** graphene, nanodevices, single-electron transistor, electrometer

## Abstract

The integrated structure of graphene single-electron transistor and nanostrip electrometer was prepared using the semiconductor fabrication process. Through the electrical performance test of the large sample number, qualified devices were selected from low-yield samples, which exhibited an obvious Coulomb blockade effect. The results show that the device can deplete the electrons in the quantum dot structure at low temperatures, thus, accurately controlling the number of electrons captured by the quantum dot. At the same time, the nanostrip electrometer coupled with the quantum dot can be used to detect the quantum dot signal, that is, the change in the number of electrons in the quantum dot, because of its quantized conductivity characteristics.

## 1. Introduction

Due to its exceptional electronic properties, graphene displays various electron transport phenomena, such as the single-electron tunneling effect [1], the anomalous quantum Hall effect [2,3], and the electron coherence effect [4]. Among them, graphene single-electron transistors (GSETs) based on single-electron tunneling and the Coulomb blockade effect can realize quantum computing functions at the nanoscale or in single-charge ultra-high sensitivity electrical quantity detection. It has wide application prospects in ultra-sensitive electrometers [5,6], single-photon detectors [7,8,9], high-density information storage [10,11], and quantum information devices [12,13].

Plenty of jobs have been performed toward GSET fabrication and electronic measurement studies. Ponomarenko et al. prepared GSET by electron beam lithography, oxygen plasma etching, and other processes [14]. The narrow graphene region between the graphene quantum dots and the source or drain electrode forms a quantum tunnel barrier. At low temperatures, the conductance of GSET shows a periodic function of the gate voltage, which is caused by the Coulomb blockade effect. When the diameter of graphene quantum dots in the device is reduced to ~15 nm, the width of the quantum barrier is only ~1 nm. By tuning the source, drain, and gate voltages, a clear Coulomb diamond image can be obtained. Even at room temperature, the device still has good switching-off performance.

Ihn et al. carried out extensive and in-depth research on GSET [15]. They fabricated GSET with a multi-gate structure and studied the energy gap in graphene nanoribbons [16], the electron–hole inversion effect of graphene quantum dots in the case of a vertical magnetic field [17], and the spin state of graphene quantum dots. Under the action of a planar magnetic field, Zeeman splitting of the spin state occurred, with a g factor of 2 [18].

The conductance of quantum point contact (QPC) is quantized. In the transport between quantum conductance platforms, it is very sensitive to the electrostatic environment, including the number of electrons n on the nearby quantum dot. This characteristic makes it possible to determine the absolute number of electrons on the quantum dot even if the electron tunneling is so weak that the current passing through the quantum dot cannot be detected. Moreover, Kouwenhoven et al. took the lead in realizing charge-sensing measurements in quantum dots by using quantum point contact (QPC) in two-dimensional electronic gas (2DEG) quantum dot devices [19]. J. Güttinger et al. performed charge detection experiments on graphene quantum dots at a low temperature of 1.7 K [5], with a 45 nm wide graphene strip as a charge detector, which is 60 nm apart from the Coulomb island with a diameter of ~200 nm. L. Lv et al. [20] and G. P. Guo et al. [21,22,23,24] have also carried out a lot of research on GSET devices. They have successfully realized the coupling of graphene quantum dots quantum bits and superconducting microwave cavity quantum data bus, which takes an important step toward the realization of integrated quantum chips.

Inspired by previous work, we fabricate an integrated structure of GSET and nanostrip electrometer based on the chemical vapor deposition (CVD) growth of graphene using semiconductor fabrication techniques, such as electron beam exposure, ultraviolet lithography, oxygen plasma etching, and electron beam evaporation. Combined with low-temperature and low-noise phase-locked amplification measurement technology, the electrometer is able to detect the change in the number of charges in the coupled quantum dot, thus, achieving sensitivity detection of a few electrons or even a single charge. Using graphene grown by CVD to design and prepare devices, the sample survival rate is not high, so we prepare a large number of devices for testing in the laboratory, but we explore the future of the mass production of devices. Perhaps, this sensing and detection technology will play a role in the research of two-dimensional material quantum information.

## 2. Method

### 2.1. Device Preparation

Figure 1A shows a scanning electron microscope (SEM) image of a graphene quantum dot device chip. The pattern of device electrodes is shown in Figure 1B, which corresponds to the electrodes in the SEM image of the quantum dot and nanostrip electrometer structure shown in Figure 1C. The device was based on ~500 μm thick doped silicon with an oxide insulating layer of ~300 nm (Figure 1D). Graphene was grown on copper foil using the chemical vapor deposition (CVD) method before being transferred and patterned by electron beam lithography (EBL) and oxygen plasma etching. Metal graphene contact resistance is one of the main limiting factors of graphene technology development in electronic devices and sensors [25]. Ohmic contact electrodes such as source electrodes, drain electrodes, gate electrodes, and nanostrip electrometers were prepared using ultraviolet lithography and evaporation processes. We used Ti/Au with a thickness of 10/300 nm to form ohmic contact with graphene. The metal evaporation rate was 0.5/3 Å/s, the sample fixture rotation speed was 8 rpm, and the metal evaporation time was 20 min. During the evaporation process, the substrate temperature rose from 27 °C to 45 °C. The back gate was used to adjust the Fermi energy level of graphene to control the electron injection ability of the device.

Graphene nanostrip showed quantized conductance due to the quantum confinement effect. We used a graphene nanostrip near quantum dots as an electrometer (Figure 1C). Similar to QPC, it can realize the function of charge sensing. The tunnel barrier of the nanostrip is determined by its size, the gate voltages, and other auxiliary mechanisms.

For the preparation of CVD graphene quantum dot devices, the process flow can be divided into the following steps: making nested marks, etching graphene mesas, depositing ohmic contact electrodes, etching quantum dot structures, preliminary screening of devices, fragmentation, bonding, and packaging. Before adhering the chip to the socket for bonding, we used a diamond knife to cut the oxide layer of the silicon substrate, applied silver glue, and cured it at 180 °C for 30 min to complete the fabrication of the back gate. Finally, in order to reduce the influence of the residual impurities on the surface of graphene, control the introduction of scattering sources, and improve the electrical performance of the device, we annealed graphene at 300 °C in a mixed atmosphere of hydrogen and argon (at flow rates of 400 and 200 sccm, respectively).

### 2.2. Device Test

Before device chip packaging, a preliminary screening at room temperature should be carried out. Under the protection of photoresist, we used a probe table to measure the resistance of the source and drain channels and measure the continuity of the test system circuit and its corresponding relationship with the device pins. For devices with source–drain channel resistance greater than 26 kΩ and in the order of 100 kΩ, the devices marked as possibly qualified can enter the next step of bonding packaging and testing. Secondly, we conducted a leakage test of the device gate. With the source and drain grounded, the leakage current of the device’s back gate and plunger gate electrodes was measured. Next, we tested the control ability of the device back gate. On the one hand, we set different back gate voltages to measure the source and drain current of the device by scanning the source and drain bias voltage. On the other hand, we set a certain source–drain bias voltage and measured the source–drain current of the device by scanning the back gate voltage to find the range of the Dirac point. Then, the control performance of the device plunger gate was tested. We set the back gate voltage in the Dirac point range, the source and drain bias voltage to certain values, and scanned the plunger gate voltage to measure the source and drain current of the device. After this, we tested the source–drain I–V characteristics of the device. We ground the device gate, adjusted the source meter, and determined the source and drain voltage range within ±10 nA of the source and drain current; then, data acquisition software was used to repeatedly scan the source leakage voltage range and collect the source leakage current data. Using the same method, we measured the I–V characteristics of the nanostrip electrometer. Finally, the charge stability diagram of the device was tested. The differential conductance of source–drain was measured by scanning the back gate and source–drain bias voltage at the same time by setting the plunger gate voltage to a certain value; alternatively, the source–drain differential conductance was measured by scanning the plunger gate and source–drain bias voltage by setting the back gate voltage in the Dirac point range.

## 3. Results

As graphene grown by CVD is transferred from copper foil to a Si substrate, its adhesion is not strong, and graphene is very vulnerable to damage or fall off during the process, thus, affecting the survival rate of samples. After etching the worktops of the three samples, we made statistics on the intactness and damage of the graphene worktops. The survival rates of the worktops were 29%, 40%, and 53%. The yield of graphene used to prepare quantum dot single-electron devices is not high, and the damage to graphene by the process is one of the important reasons. Using paraffin instead of polymer for transfer can effectively reduce wrinkles and polymer residues and reduce defects and damage [26]. In addition, in the process, acetone immersion and ultrasonic cleaning may also cause pollution or damage to the surface of graphene, which needs to be improved.

The integrated device of graphene quantum dots and nanostrip prepared based on the semiconductor process was placed in the He^3^ closed-cycle cryostat (Cryo Industries of America, Inc., Manchester, NH 03103, USA) shown in Figure 2A for testing. The chip was fixed on the socket with silver glue (Figure 2B), and the pins correspond to the individual device electrodes. The core structure of the device is a graphene quantum dot, nanostrip, and other structures obtained by semiconductor processes, such as oxygen plasma etching (Figure 2C). Figure 2D,E show SEM pictures of one single quantum dot device and one integrated device of quantum dot and nanostrip, respectively.

### 3.1. Source–Drain Channel Resistance and Back Gate Leakage Test

As shown in Figure 3A, the source–drain channel resistance of the device measured by the four-probe method at room temperature is about 37.5 kΩ, which is greater than the quantum resistance (~26 kΩ) and less than 1 MΩ, indicating that the tunnel barrier resistance of the device is in a reasonable range. In the device structure design, the ~500 μm thick Si sheet is the substrate, and there is a 300 nm oxide layer on the surface. The substrate silicon is the back gate, and the oxide layer is the back gate insulation layer. The back gate leakage characteristic curve at ~7.7 K is shown in Figure 3B. The insulation layer resistance of the back gate electrode is 6.0 × 10^9^ Ω. This indicates that the back gate insulating layer of the device has good performance and no leakage occurs.

### 3.2. Test of Regulation Capability of Back Gate and Plunger Gate

The ability of gate regulation is very important for quantum dot single-electron devices. Coulomb blocking can be removed by changing the gate voltage. For graphene quantum dot devices, the back gate voltage can effectively adjust the Fermi level shift in the entire graphene nanostructure, while the plunger gate electrode can be used to locally adjust the chemical potential level in the quantum dot [27]. The performance of the device can be checked by testing the regulation ability of the gates.

As shown in Figure 4A, the back gate is set with different voltages *V*_bg_, the source–drain electrode is connected to the source meter, the source–drain bias voltage *V*_ds_ is scanned, and the source–drain current *I*_ds_ is measured at 250 K, as shown in Figure 4B. The results show that the slope of the source–drain I–V characteristic curve of the device changes significantly with the change in the back gate voltage *V*_bg_, indicating that the back gate voltage effectively regulates the graphene energy level and changes the carrier concentration of the graphene channel. When *V*_bg_ is in the range of 0 V to 40 V, the equivalent resistance of the source–drain channel increases with the increase in voltage; however, as *V*_bg_ changes from 40 V to 60 V, the source–drain channel equivalent resistance decreases. This is the result of the back gate regulating the potential barrier of graphene nanoribbons and the discrete energy levels of graphene quantum dots. It shows that the back gate voltage changes the carrier concentration of the graphene channel, and the Dirac point is near 40 V.

Then, we set the source–drain bias voltage as the fixed value *V*_ds_ = 75 mV, the current preamplification factor as 10^−9^ A/V, and the rise time as 300 ms. Under a temperature of ~8K, the regulating curve of the device back gate to the source–drain current can be obtained, as shown in Figure 4C.

The results show that the back gate can fully adjust the Fermi surface of graphene in the device and change the carrier concentration involved in transport. We adjusted the transmission from the hole (left side) to the electronic state. The back gate voltage can inhibit transmission in the range of 19 mV < *V*_bg_ < 46 mV (Δ*V*_bg_ ≈ 27 mV). In the region of the restraining current, there are a lot of sharp resonances because there is an effective energy gap in the bias direction within the transmission gap of the back gate voltage. Outside the suppression area, the current peaks a~e, as shown in Figure 4C, are the Coulomb oscillation peaks obtained by back gate regulation.

The edge configuration of graphene quantum dots and nanoribbons has a great influence on their properties. Uncontrolled factors, such as the size of graphene quantum structures and the location and concentration of surface defects (vacancies and impurities), will also affect the energy gap. Because graphene quantum dots and nanoribbon structures have different quantum confinement effects at different positions, as well as the local states caused by rough edges [16], the back gate electric field will have different effects on the discrete energy levels of quantum dots and the potential barriers of nanoribbon structures at the same time (Figure 4E). This edge-induced disordered potential energy will affect the transport of graphene nanoribbons. With the increase in the scanning range of the back gate voltage, the discrete energy levels of the quantum dots will be located in the source–drain bias window. At this time, the equivalent resistance of the source–drain channel decreases, the current increases, and Coulomb oscillation occurs.

In the range of −70 mV to 100 mV, we set the source–drain bias voltage *V*_ds_ to a series of determined values at equal intervals (10 mV), scanned the back gate voltage *V*_bg_, and measured the current *I*_ds_ passing through the quantum dot, and we obtained the results shown in Figure 4D under a temperature of ~8 K. The I–V characteristic curve shows the rudiment of the Coulomb diamond.

As shown in Figure 5A, the source–drain bias voltage is set to *V*_ds_ = −7.4 mV, the back gate electrode is grounded, and the source–drain current *I*_ds_ is measured by scanning the plunger gate voltage *V*_pg_ at ~6.9 K to obtain the curve shown in Figure 5B. The repeated test results show that the device has obvious Coulomb oscillation characteristics under plunger gate control. The current peak indicates that there is an electrochemical potential level corresponding to the continuous ground state transport, which is located between the source and drain and generates a single-electron tunneling current. The wave trough indicates that the Coulomb blocking effect has occurred, and the number of electrons on the quantum dot is stable. By adjusting the plunger gate voltage, the current can move from one trough to the next so that the number of electrons on the quantum dot can be accurately controlled.

### 3.3. Source–Drain I–V Characteristic Test

Coulomb repulsion between electrons on a quantum dot causes an external electron to consume energy when entering the quantum dot. This phenomenon is called the Coulomb blockade effect. When the external magnetic field is zero, for a quantum dot system containing a certain number of electrons in an equilibrium state, the electron transmission can only occur when the electron level in the quantum dot corresponding to the transport is in the source–drain bias window. If this condition is not satisfied, the number of electrons on the quantum dot will be fixed. Coulomb blockade can be removed by changing the source–drain voltage.

The electrical circuit of the device measurement is shown in Figure 6A. The resistance *R*_1_ = *R*_2_ = 1 MΩ on the source–drain channel plays the role of current-limiting protection and voltage division. The resistance *R*_3_ = 1 kΩ, *R*_2_, and *R*_3_ form a voltage divider to improve the voltage resolution and achieve the effect of improving the test signal-to-noise ratio. Assuming that the source–drain channel resistance *R*_0_~10^5^ Ω (*R*_0_ >> *R*_3_), then *V*_a_ = *R*_3_*V*_ds_/(*R*_2_ + *R*_3_). Since *R*_2_ >> *R*_3_, the voltage of the input device can be obtained as *V*_a_ ≈ *V*_ds_/*R*_2_ = *V*_ds_/1000, thus, realizing the 1000-time subdivision of the DC bias voltage.

The back gate and plunger gate are grounded. By scanning the source–drain voltage *V*_ds_ and simultaneously measuring the current passing through the quantum dot *I*_ds_, the curve shown in Figure 6B can be obtained. The results show that the device shows an obvious Coulomb blocking effect at a low temperature of ~8.4 K. It can be obtained that the voltage width of the blocking area near *V*_ds_ = 0 V is about 103 mV. It is believed that the charging energy *E*_C_ of the device Coulomb island is 103 meV, which is far greater than the thermal energy at 8.4 K (≈0.72 meV). The total capacitance of the Coulomb island is *e*^2^/*E*_C_ ≈ 1.55 aF. 

The constant current part of the current step shows that the Coulomb blockade effect occurs and the number of electrons on the quantum dot is stable. However, the quantum dot has an electrochemical potential level corresponding to the continuous ground state transport, which is located between the source and drain and can continuously generate a single-electron tunneling current. The rising section of the current step shows that by adjusting the source–drain voltage, the number of electrochemical potential energy levels of the quantum dot at the source–drain bias window changes, the number of electrons on the quantum dot changes, and the current moves from one step to the next. Based on this, the number of electrons on the quantum dot can be precisely controlled.

### 3.4. Charge Stability Diagram Test

The finite size of quantum dots in three dimensions will affect the electronic dynamics, resulting in quantum effects, thus, forming the discrete energy spectrum of quantum dots. Therefore, Coulomb blocking, Coulomb oscillation, and Coulomb prisms can be measured and single-electron states can be prepared in quantum dot single-electron devices. When the gate voltage and the source–drain voltage are changed at the same time, the Coulomb step effect and the Coulomb oscillation effect work together; thus, we can observe the Coulomb diamond of the conductance of the quantum dot device. The diamond area indicates that the conductance is zero, and the charge cannot tunnel through the dot. Each diamond corresponds to a stable charge configuration of several occupied electrons on the quantum dot. The difference in the number of charges on the quantum dots represented by adjacent diamonds is one.

Using the standard electrical transport measurement method of quantum dot single-electron device shown in Figure 7A, the results shown in Figure 7B can be obtained. The device was placed in a vacuum cryogenic dewar, and the electrical test was carried out only after the system temperature dropped to 2.7 K. The device gate was connected to the DC voltage source meter through a 1 MΩ protection resistor, and the scanning range was 10 V to 20 V; the AC voltage reference signal (0.265 V rms at 177 Hz) given by the lock-in amplifier (Stanford Research Systems, Inc., SR830, Sunnyvale, California 94089, USA) and DC voltage (−28 mV ↔ +28 mV) given by the source meter was added to the source end of the device, and the current flows out of the drain and enters the current preamplifier (amplification factor, 10^−8^ A/V; rise time, 0.01 ms), which is converted into a voltage signal and enters the phase-locked amplifier for correlation operation (time constant, 500 ms) and, finally, is read by a digital multimeter.

In the circuit shown in Figure 7A, the capacitance *C* = 1 μF acts as a DC isolation. According to the Kirchhoff current conservation law, the node voltage *V*_b_ meets
(1)Vb=R5||(1jwC+R3||Rdev)R4+R5||(1jwC+R3||Rdev)Vout

The resistance *R*_5_ = 1 Ω, the AC signal amplitude of the phase-locked amplifier is 1 V, and the frequency is 137 Hz. Since *R*_2_, *R*_4_, 1/(jw*C*) >> *R*_3_, *R*_5_, Equation (1) can be simplified as:(2)$$V_{b}\approx \frac{R_{5}||\left(1/jwC+R_{3}\right)}{R_{4}+R_{5}||\left(1/jwC+R_{3}\right)}V_{out}\approx \frac{R_{5}}{R_{4}+R_{5}}V_{out}\approx \frac{R_{5}}{R_{4}}V_{out}$$

Thus, the voltage divider composed of *R*_4_ and *R*_5_ realizes the purpose of obtaining a small signal from the reference signal of the lock-in amplifier and inputting it to the device.

Further analysis shows that the current at node *V*_a_ meets the following formula:(3)$$\frac{V_{ds}-V_{a}}{R_{2}}+jwC\left(V_{b}-V_{a}\right)=\frac{V_{a}}{R_{3}}+\frac{V_{a}}{R_{dev}}\approx \frac{V_{a}}{R_{3}}$$

Calculated as *V*_a_ ≈ (*V*_ds_ + j1.37 × 10^−4^ *V*_out_)/(1001 + j0.137), which realizes the addition and subdivision of the DC bias voltage and the AC signal.

It can be seen from Figure 7B that the plunger gate voltage variation in the adjacent rhombus is Δ*V*_pg_ ≈ 1.78 V; thus, the gate capacitance is *C*_pg_ ≈ 0.09 aF. The slope of the two edges of the Coulomb diamond is *k*_d_ = −*C*_pg_/*C*_d_ = −0.027 and *k*_s_ = *C*_pg_/(*C*_pg_ + *C*_s_) = 0.045, so *C*_d_ = 3.33 aF and *C*_s_ = 1.91 aF. As *C*_s_ ≠ *C*_d_ is asymmetric, it shows that the source and drain tunneling resistances are different, and their junction capacitances are different, so the Coulomb diamond is inclined.

When the source–drain voltage is large enough, more energy levels in the quantum dot will participate in electron tunneling, and more excited state energy levels will be located in the bias window, which will lead to the transition from single-electron tunneling to multi-electron tunneling. In Figure 7B, we can observe the excited state energy spectrum of the single-electron transistor. The absence of charge transfer on the left side of Figure 7B shows that the electrons in the quantum dot structure are completely depleted, that is, the number of free transfer charges on the quantum dot is zero. With this area as a reference, the absolute number of electrons in the area of interest can be known. If the gate voltage and source–drain bias voltage are changed, the number of electrons on the quantum dot will change, and the conductance can go from one diamond to the next. The gate voltage changes by Δ*V*_pg_ = |*e*|/*C*_pg_, so the number of electrons on the quantum dot can be precisely controlled.

Figure 7C shows that the nanostrip exhibits Coulomb-blocking transmission characteristics because the fluctuation in the strip edge causes the change in tunneling ability, making the nanostrip behave like quantum dots in series.

The integrated structures of the quantum dot and nanostrip are jointly tested at 310 mK. The results shown in Figure 8 indicate that the device has a sensitive charge detection capability. The single-electron transport in the quantum dot will certainly cause a change in the number of charges in the quantum dot. The electrometer is very sensitive to the electrostatic potential in its neighborhood and can measure the change in the electrostatic potential caused by the change in the number of single charges in the quantum dot. This is because the nanostrip electrometer has quantized conductivity due to the quantum confinement effect, and its function is similar to that of quantum point contact. The transport between adjacent conductive steps is extremely sensitive to the electrostatic environment of its neighborhood, thus, realizing the single-charge ultra-high sensitivity electricity detection.

In these measurements, the back gate voltage is set to 5.0 V, where the quantum dot is nearly electrically neutral and within the transmission gap of the electrometer. We manipulate the electrometer under the condition that strong resonant interaction can be obtained between electrons and energy levels in the nanostrip so as to use the steep slope of conductance regulated by the gate to detect a single-charge event of the quantum dot. At this time, the detection sensitivity is the highest.

Because the low-energy particles in graphene are Dirac fermions, which have the Klein tunneling effect, graphene is usually patterned to nanoribbon to form a potential barrier that constrains electrons in order to obtain quantum dot structures. The nanoribbon should not be too wide; otherwise, the resistance is small and cannot act as a potential barrier; the nanoribbon should not be too narrow; otherwise, the resistance is small and the barrier has no tunneling function. At the same time, the strip length should not be too long; otherwise, an effect similar to that of multiple quantum dots in a series may be formed due to irregular edges.

The integration of the graphene quantum dot structure and the nanostrip electrometer can be realized by using EBL and oxygen plasma etching technology. By reasonably designing the length and width of graphene nanoribbons connecting graphene quantum dots with source and drain electronic libraries, the tunnel barrier between the quantum dots and source and drain electrodes can be controlled to ensure that the conduction impedance between source and drain is in the range of 50~500 kΩ at room temperature. The single-electron transport properties of the quantum-dot- and nanostrip-integrated structure and the gate-controlled conductivity of the electrometer were measured, and the sensitivity of the nanostrip electrometer to the electrostatic environment was verified.

## 4. Conclusions

In summary, we used electron beam exposure and reactive ion etching (oxygen flow, 50 SCCM; RF power, 500 W; pre-stage power, 200 W; air pressure, 100 mTorr; and etching time, 20 s) and other techniques to pattern CVD graphene, obtained graphene quantum dots with a diameter of ~100 nm, and connected to the source and drain electrodes through a narrow nanoribbon with a width of ~40 nm to prepare back gate graphene quantum dot devices. We used plasma cleaning, high-temperature annealing, and other measures to effectively reduce the interaction between graphene and the substrate, remove the residue on the surface of graphene, and control the introduction of impurity scattering sources to ensure the performance of the device.

The electrical properties of graphene quantum dot devices were tested, and the precise control of the number of electrons captured by the quantum dots was realized. In the charge stability diagram, the absence of charge transfer indicates that the electrons in the quantum dot structure are completely depleted. It can be seen that by adjusting the gate voltage and the source–drain bias voltage, the number of electrons on the quantum dot can be accurately controlled, and the preparation of single-electron states in the quantum dot can be realized. The electron transport can change the number of charges in the quantum dot and affect the electrostatic potential of the electrometer. The conductivity-quantized nano-strip electrometer is very sensitive to the electrostatic environment in the neighborhood and realizes sensitive charge detection.

The limit of traditional current measurement is ~10 fA, and the lower limit of the corresponding tunneling probability is ~100 kHz. To measure the quantum dot signal with the tunneling probability of the electronic library of less than 100 kHz, using the electrometer- and quantum-dot-integrated structure is an optional method. The electrometer is integrated near the quantum dot as a charge detection mechanism, which can detect the number of electrons in the quantum dot without affecting the electron transport in the quantum dot and has high measurement sensitivity. It is expected that this technology will play an important role in the research of quantum information.

## Figures and Tables

**Figure 1 nanomaterials-13-00889-f001:**
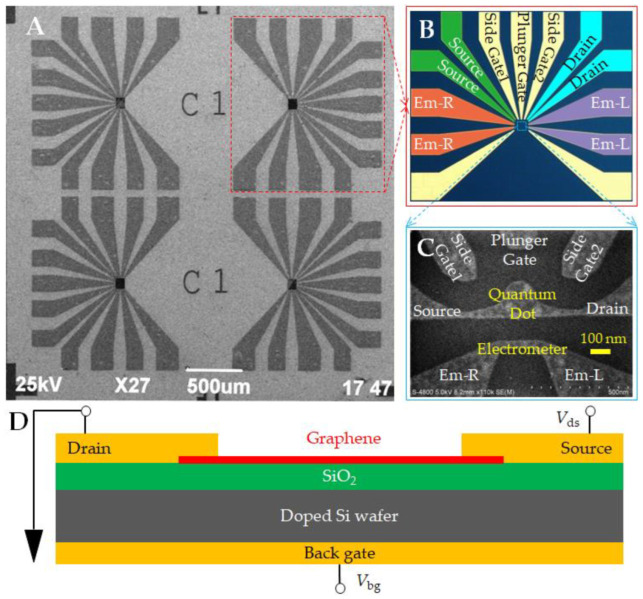
(**A**) SEM image of graphene quantum dot device chip; (**B**) electrode arrangement; (**C**) SEM image of quantum dot and nanostrip electrometer structure; and (**D**) device cross-section diagram. The source and drain are connected with the quantum dot via graphene nanoribbon.

**Figure 2 nanomaterials-13-00889-f002:**
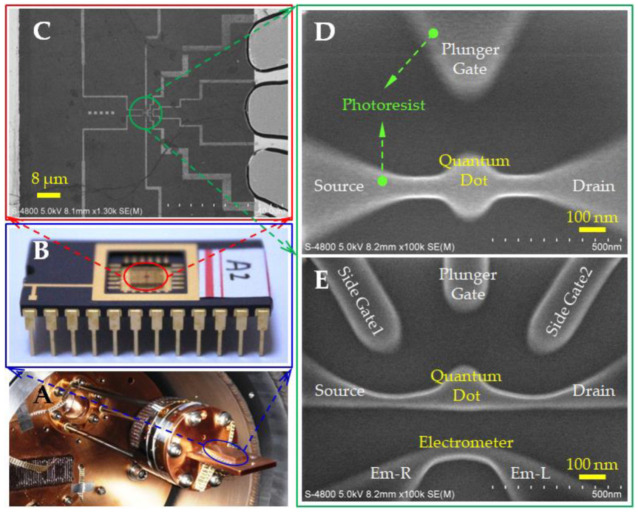
(**A**) He^3^ closed-cycle cryostat cold head for device test; (**B**) the chip carrier matched with the cold head. After the chip is fixed on the tube base, each electrode of the device is led to the corresponding pin by means of gold wire ball welding; (**C**) SEM images of graphene functional areas in the device; (**D**,**E**) SEM image of photoresist mask on graphene mesa before etching quantum dot structure.

**Figure 3 nanomaterials-13-00889-f003:**
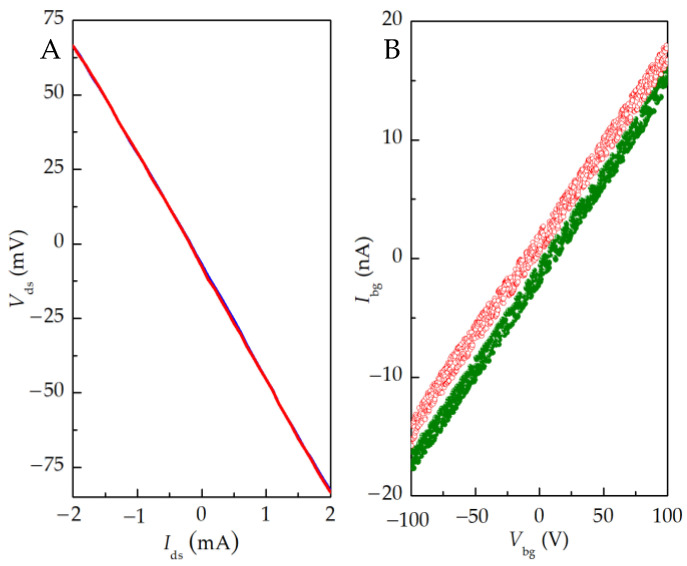
(**A**) I–V curve of device source–drain channel measured by four-probe method. The device source is connected to the source meter YOKOGAWA GS200 and a digital multimeter Agilent 34461A, and the back gate and drain are grounded. At room temperature, two points A–B on the outside of the source–drain channel are connected to a constant current source, and the voltage of two points a–b on the inside is measured. (**B**) Leakage characteristics of device back gate. The source–drain was grounded, the leakage current of the back gate electrode was measured with a Keithley 2410 at ~7.7 K. The red and green curves represent the leakage current obtained when the back gate voltage was scanned from −100 V to +100 V and +100 V to −100 V. The curves with different colors represent the data obtained by different scans, with good repeatability.

**Figure 4 nanomaterials-13-00889-f004:**
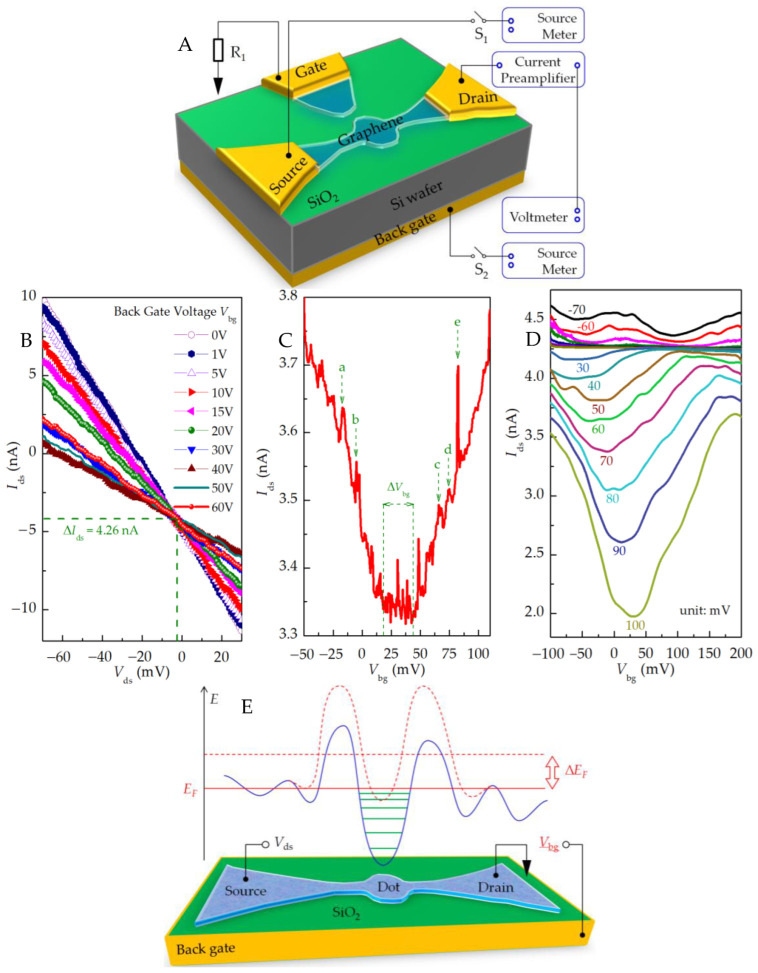
(**A**) Circuit diagram of back gate regulation test. The source–drain current *I*_ds_ is measured with a current preamplifier (DL Instrument 1211; amplification factor, 10^−5^ A/V; rise time, 300 ms) and a digital multimeter. (**B**) Source–drain I–V characteristic curves under different back gate voltages intersect at *V*_ds_ ≈ −2 mV, and the source–drain current has a bias current of ~4.26 nA, which is introduced by the preamplifier test circuit. Back gate regulation characteristics under (**C**) a certain source–drain bias and (**D**) different source–drain bias voltages. (**E**) Diagram of potential energy along the graphene source–drain channel. The blue solid line represents the potential energy curve of graphene source and drain channel. The movement and change of different positions under the control of the back grid are different, and may finally have the shape of red dotted line.

**Figure 5 nanomaterials-13-00889-f005:**
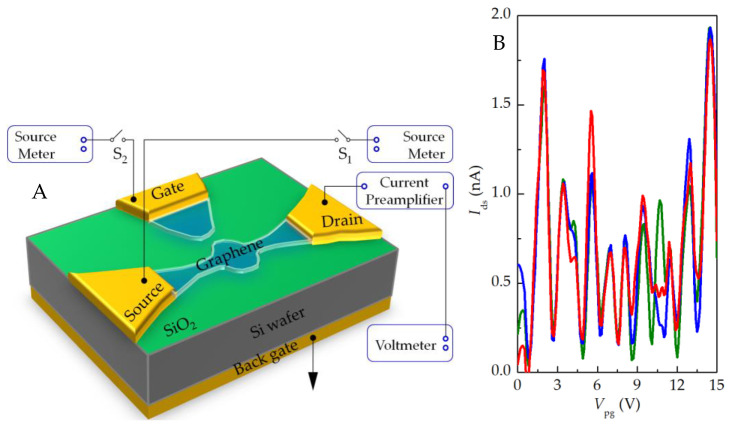
(**A**) Circuit diagram of plunger gate regulation test; (**B**) plunger gate regulation curve under certain source–drain bias voltage. The different color curves of red, green and blue represent the test curves obtained by different scanning, and the data repeatability is good.

**Figure 6 nanomaterials-13-00889-f006:**
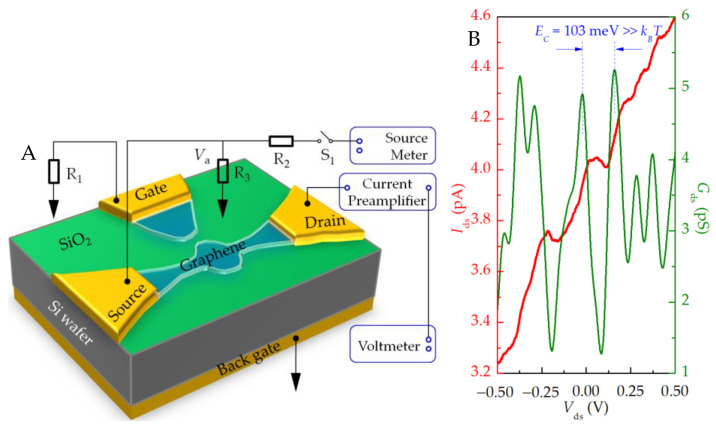
(**A**) Electrical connection of device test. In the measurement, the current preamplification factor is 10^−9^ A/V, and the rise time is 300 ms; (**B**) I–V characteristic test curve of source–drain. The red curve corresponds to the left ordinate, indicating the source leakage current; The green curve corresponds to the ordinate on the right, indicating the differential conductance.

**Figure 7 nanomaterials-13-00889-f007:**
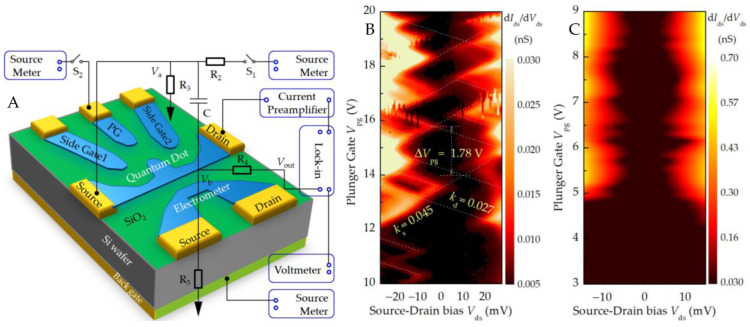
(**A**) AC phase-locked differential conductance current amplification test circuit. Differential conductance diagram of (**B**) the single-electron transistor and (**C**) nanostrip integrated with quantum dots measured at 2.7 K liquid helium temperature.

**Figure 8 nanomaterials-13-00889-f008:**
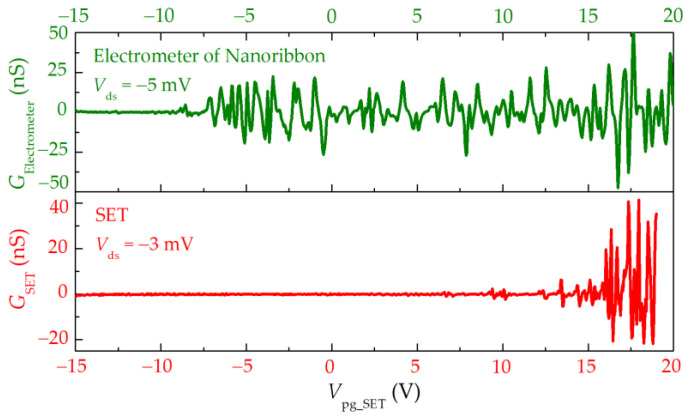
Comparison diagram of conductance curve of quantum-dot–nanostrip-integrated device.

## Data Availability

All data, models, and code generated or used during the study appear in the submitted article. They are available from the author by request (J.Y. Fang).

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
