# Peer review of "Single-Electron Transport and Detection of Graphene Quantum Dots"

_nanomaterials, 2023, doi:10.3390/nano13050889_

Round 1
Reviewer 1 Report
The authors present a design of a GSET coupled to a nano strip electrometer. They perform several test measurements to show the functioning of the design. However, a pure design study is not in the scope of the Journal nanomaterials.
Does this design work for more than one GSET and one electrometer?
How can the design be modified to be integrated into a circuit that enables the manipulation and reading of the GSET states?
What is the scientific information from the manuscript? The authors might revise the manuscript to present clear results rather than succcessful test procedures which apply for only one single device so far.
In addition, two different statements about the origin of the graphene are made. On page 2 lines 72-74 it is written: " Graphene is grown on the substrate by CVD method before patterend...". In contrast, on page 4, line 124-126 it's said: " As graphene grwon by CVD is transferred from Cu foil to Si substrate...". Please give a clear description on the fabrication and on the origin of the graphene used in this study. If the graphene is transferred from Cu foil, from which company was the graphene purchased?
Author Response
Dear Reviewer 1# and editor,
Thank you very much for your attention and kind comments on our paper. In the attachment, we have made a detailed response to your comments and suggestions one by one.
We sincerely hope that this manuscript will eventually be accepted and published in the journal Nanomaterials.
Kind regards, 
Jingyue Fang
2023-2-22

Reviewer 2 Report
In this paper the authors apply standard semiconductor processing techniques to fabricate graphene single-electron transistors with integrated nano strip electrometer. The electrometer can detect the change of charge in the coupled quantum dot and achieve sensitivity detection of a few electrons or even of a single electron.
The study is interesting from both a fundamental and application standpoint. The paper reports a well-performed and analyzed experimental work.
The paper is not easy to understand in some parts but can become suitable for publication after a revision:
“Inspired by previous work, we fabricated integrated structure of GSET and nano strip electrometer using semiconductor fabrication …” The authors should clearly state which is the real novelty, if any, of their paper with respect to the previous works. As presented in the introduction, it seems that this study only confirms what was already reported for instance by Ponomarenko et al.
“Ohmic contact electrodes such as source electrode, drain electrode, gate electrode and nano strip electrometer are prepared by ultraviolet lithography and evaporation processes.” Specify which metal is used to form ohmic contacts. Low-resistance ohmic contacts are quite important in graphene transistors (see for instance https://doi.org/10.1088/2632-959X/ab7055). Some discussion can be added on this point.
“The yield of graphene used to prepare quantum dot single electron devices is not high, and the process damage of graphene is one of the important reasons.” If possible, the authors could suggest any strategies to overcome this issue and increase the yield.
“As shown in Figure 3A, the source drain channel resistance of the device measured 146 by the four probe method in room temperature is about 37.5 kΩ, which is greater than the 147 quantum resistance (~26 kΩ) and less than 1 MΩ, indicating that the tunnel barrier resistance of the device is in a reasonable range.” Clarify which tunnel barrier is referred to here. Till now it is not very clear where are and which is the origin of the tunnel barriers needed to realize a single electron transistor.
Figures 3a and b are a little confusing. I suggest plotting the voltage always on the x-axis. State what are the red and green curves in figure B? The straight lines in figure B are rather unusual for a gate leakage. I would expect some nonlinear curve, considering that this should be the current through an oxide. Also, I see that a back gate voltage of up to 100 V is applied. Such a high voltage can easily cause the breakdown of the 300 nm gate oxide. Some clarification is needed here.
“For graphene quantum dot devices, the back gate voltage can effectively adjust the Fermi level shift of the entire graphene nanostructure, while the plunger gate electrode can be used to locally adjust the chemical potential level in the quantum dot.” The effect of a graphene side gate on a graphene channel is reported in this paper https://doi.org/10.1063/1.4958618 which can be cited here. Which is the role of the lateral gates? Are they used anyway?
The plot of figure 4E might be not so immediate to understand. Some additional explanations in the text could be appropriate.
The curves in figure 5B are measured at 8K. I wonder if the authors have performed the same measurements at higher temperatures. Which is the result of a similar measurement at room temperature?
Author Response
Dear Reviewer 2# and editor,
Thank you very much for your attention and kind comments on our paper.
These suggestions are crucial to improve the quality of the article. According to your suggestion, we have explained or revised the problems in the manuscript. In the attachment, we have made a detailed response to your comments and suggestions one by one.
We sincerely hope that this manuscript will eventually be accepted and published in the journal Nanomaterials.
Kind regards, 
Jingyue Fang
2023-2-22

Round 2
Reviewer 1 Report
Thanks to the authors for their comments, clarifications and revision.
Reviewer 2 Report
The authors have satisfactorily addressed all the reviewers' comments. The paper can be accepted.